# OpenReview forum: "On the Impossibility of Retrain Equivalence in Machine Unlearning"
_ICLR.cc/2026/Conference — Submitted to ICLR 2026_

### Official Review · Reviewer_S5jh · 2025-10-29

**Soundness:** 3
**Presentation:** 1
**Contribution:** 2
**Rating:** 2
**Confidence:** 4

**Summary:**

The paper considers the difficulty of unlearning with “local” algorithm, i.e., those that only use the model weights and the forget set. They conduct experiments on LLMs and note the ordering of the training data can make such unlearning algorithms significantly less effective , and provide some theory in the case of linear regression. They conclude that further work is needed in formulating this type of unlearning.

The authors seem unaware of past work which had already shown this formulation for unlearning is ill-defined; a dataset without the forget set can give the same model as training with the dataset (with some probability over the training orderings), e.g., Thudi., et al [1]. Note at the time “approximate unlearning” only really consisted of “local” unlearning algorithms. This led to further research into forgeability, and analysis of the closely related notion of per-instance privacy. It also has been one of the primary reasons for stronger algorithmic definitions of unlearning (which use the training algorithm and retain dataset) and disclosures when using weaker definitions, e.g., see the ToFU benchmark [2].

In the context of this literature, this paper makes further observations on the nature and prevalence of forgeability. However, this paper does contribute to the existence of this phenomenon, which was already proved in general for mini-batch SGD training algorithms [1] using an even stronger metric (i.e., the model weights are the same which implies predictions are the same). I recommend this blog post for a more intuitive understanding of forging https://ai.stanford.edu/~kzliu/blog/unlearning (which is described in terms of verifying unlearning), and the work of Kong et al., [3] which recaps the results of [1] a bit more cleanly.

In the questions I make some suggestion for how the authors might change their presentation given this; while I do not think the original claims are valid, the paper can claim to add to existing observations about this unlearning setting.

[1] Thudi, Anvith, et al. "On the necessity of auditable algorithmic definitions for machine unlearning." 31st USENIX security symposium (USENIX Security 22). 2022.

[2] Maini, Pratyush, et al. "Tofu: A task of fictitious unlearning for llms." arXiv preprint arXiv:2401.06121 (2024).

[3] Kong, Zhifeng, Amrita Roy Chowdhury, and Kamalika Chaudhuri. "Forgeability and membership inference attacks." Proceedings of the 15th ACM workshop on artificial intelligence and security. 2022.

**Strengths:**

1) The linear regression theorem adds to theory around quantifying notions of forgeability.

2) Experimental results quantifying the path dependence of various local unlearning algorithms seem strong and valuable to the literature on unlearning difficulty.

**Weaknesses:**

1) The theory, while interesting, is still focused on linear regression where efficient effective unlearning is not a problem, e.g., Guo et al., [4].

2) Moreover, bounds for the difficulty of unlearning when using SGD like algorithms (even when applied to deep neural networks) already exists, Sephavand et al., [5]; the argument deep learning is mysterious so we work with linear regression seems ill-made given the progress in the unlearning literature.

3) On the observation of path dependence (i.e., one starts with different initial models), old work (before even the unlearning community's observation) had already discussed the high variance in models between data orderings and how this can be exploited [6].


[4] Guo, Chuan, et al. "Certified data removal from machine learning models." arXiv preprint arXiv:1911.03030 (2019).

[5] Sepahvand, N.M., Thudi, A., Isik, B., Bhattacharyya, A., Papernot, N., Triantafillou, E., Roy, D.M., Dziugaite, G.K.. (2025). Leveraging Per-Instance Privacy for Machine Unlearning. Proceedings of the 42nd International Conference on Machine Learning

[6] Shumailov, Ilia, et al. "Manipulating sgd with data ordering attacks." Advances in Neural Information Processing Systems 34 (2021): 18021-18032.

**Questions:**

Below I give some suggestion for how the authors might re-present their work, which I think can help them be clear about what their contribution to the literature is. I think with such changes to the presentation this paper can be valuable to the literature on unlearning, and will be happy to reconsider the score. However, the paper misses the mark right now given it lacks context with existing results.

Suggestions:

1) While the phenomenon that unlearning is dependent on training trajectory is known in various contexts (which can lead to it being ill-defined), it seems a concrete contribution of this paper is that it measures this dependence (often called forging) for LLM fine-tuning. I suggest the authors no longer claim to discover the former, and instead discuss past work on this phenomenon and more broadly unlearning difficulty. They can then just focus on their contribution to measuring this dependence in LLM fine-tuning, which is already mentioned throughout the paper (but often after claiming the former).


2) Given this, the description of the contribution of the theoretical result could be rephrased to emphasize that it quantifies how specific orderings diverge with continual “local” unlearning; the exponential divergence seems novel to the unlearning (difficulty) literature, and is already what is described in the theorem. One just needs to rephrase the summary of this result (in the context of past results) in the introduction and elsewhere in the paper.

3) Similarly the empirical section seems to be correct on its own, and just how these results are interpreted in the context of past results needs to be made clearer. For example, as far as I know the observation about the recency phenomenon seems novel, and to me is something worthy of a paper; one just needs to not claim to be the first to observe that machine unlearning depends on data ordering, and instead only make the more fine-grained claims of what dependencies were specifically observed.

---

> ### Author Response · Authors · 2025-12-03
>
> We thank the reviewer for recognizing the novelty and importance of (1) our theoretical results on the exponential divergence speed of unlearning, and (2) our empirical findings on path-dependent divergence of unlearning.
>
> The reviewer pointed to prior work on forgeability as being very similar. We  plan to cite it in the final draft, but wish to point out the following.
>
> **(1) (Summary) Relationship between our work and Thudi et al.**
>
> Thudi et al. complements our main idea on Retrain Equivalence being ill-posed, but gives a very different reason to be suspicious of Retrain Equivalence.  We politely **disagree** with the reviewer’s criticism that our work’s contribution is in “mak[ing] further observations on the nature and prevalence of forgeability”. Our paper is not about forgeability at all.
>
> **Context**: Thudi et al.  defined the following concepts:
>
> - Privacy guarantee: a model has not seen the forget data
> - Forgeability: A model with the same or arbitrarily similar model parameters can be obtained from a different dataset that does not contain the forget data.
>
>     By proving forgeability attacks can occur, Thudi et al. shows that “having similar parameters as the retrained model” does not prove that a particular datapoint wasn’t used, rendering approximate unlearning an ill-posed definition. Therefore even a model identical to the retrained-from-scratch ideal can’t deliver privacy guarantees.
>
>
> We want to make the following clarifications:
>
> 1. **Our results are not about forgeability**.
>
>     Thudi et al. (and Kong et al.) argues that, since two different datasets can yield the same model parameters, even a model identical to the Retrained Equivalence model is not guaranteed to be private.
>
>     Our work shows that it is generally impossible to obtain model behavior (thus parameters) close to the Retrain Equivalence in multi-stage training for *general datasets, without adversary attacks*. These are very different concepts.
>
> 2. **Our results give a practical, general reason why RE might be ill-posed**.
>
>     Forgeability is not a practical assumption in real life. As Suliman et al.[1] pointed out, it is highly unlikely that general datasets are forgeable to each other. Even adversarial forgeability attacks are not as successful as previously claimed: Suliman et al. used the greedy-search attack (recommended by Thudi et al. and Kong et al.), to find that forgeability attacks introduce errors of multiple orders of magnitudes higher than benign noise on MNIST and CIFAR10 — making forgeabiilty attacks easily detectable and thus an open question in practice.
>
>     Therefore forgeability only gives a *corner case, theoretical* argument on why approximate unlearning is ill-defined, and there’s currently no algorithm to implement this attack in practice.
>
>     On the other hand, our results hold with LLM post-training pipelines with widely used datasets. By showing that efficient, gradient-based unlearning methods can’t achieve Retrain Equivalence without access to training history, our work gives practical implications on unlearning design choices and evaluation protocals.
>
> 3. **Our metric over model behavior (instead of parameter space) is not weaker than parameter space.**
>
>     The reviewer incorrectly claims that our theoretical and empirical results are over a weaker metric of model prediction (in contrast to model parameters). The reviewer correctly pointed out that identical model weights imply identical model predictions. It logically follows that divergent model predictions imply divergent model weights, not vice versa.
>
>     When proving an algorithm achieves RE, parameter space metric is stronger; when proving an algorithm can’t achieve RE, prediction space metric is stronger.
>
>     The reviewer can see our response to Reviewer q9sZ for a walkthrough of why our results imply divergence over model parameter space using the more widely-adopted $(\epsilon,\delta)$-unlearning definition.
>
>
> [1]. Suliman, Mohamed, et al. "Data forging is harder than you think." *Private ML @ ICLR*. 2024.

---

> > ### Author Response · Authors · 2025-12-03
> >
> > **(2) (Weakness 1) Linear model and Relationship to Guo et al.**
> >
> > The reviewer claims that “compuationally efficient and effective unlearning is not a problem”, perhaps due to a misunderstanding of how our work defined “computationally efficient” algorithms(See the paragraph above Definition 2.2). We have clarified this in the Related Works section, where we already discussed the phenomenal work by Guo et al.
> >
> > The current work studies unlearning algorithms whose runtime only scale with the forget dataset size. This choice is due to practical concerns in scalability for large-scale neural networks. Guo et al. has specificly mentioned that the runtime scales with $O(d^2 n)$, where $d$ is the dimension of model parameters and $n-1$ is the retained dataset size.
> >
> > Despite their immense importance, certified unlearning methods whose runtime scale with other factors than forget dataset size is outside of the scope of our work.
> >
> > **(3) (Weakness 2) Relationship to existing bound on hardness of unlearning**
> >
> > Our goal is different and more structural: to isolate and quantify the specific mechanism of path-dependent divergence under multi-stage training and local unlearning on the forget set only. For this, we use an overparameterized linear model because it is simple enough to admit an exact analysis of stage-dependent unlearning dynamics.
> >
> > **(4) (Weakness 3) Relationship to Shumailov et al.**
> >
> > We appreciate the reviewer’s suggestion and have incorporated discussion of Shumailov et al. in our revised Related Works section. Shumailov introduced the setting where an adversery can re-order i.i.d. training batches to poison or slow-down the learning process of a neural network. We show that a random and adversery-free reordering of training stages can already induce exponentially-fast divergence in unlearning outcome. The results of the current work and Shumailov et al. complement each other in understanding the importance of training trajectories, albeit in different training settings.
> >
> > **(5) (Suggestions) Expand discussion of prior work**
> >
> > We appreciate the reviewer’s suggestion to discuss more past work on (1) hardness of unlearning (e.g. Thudi et al,, Kong et al.) and (2) path-dependent of learning (Shumailov et al.). We have revised the Related Works section.

---

### Official Review · Reviewer_QvRz · 2025-10-29

**Soundness:** 3
**Presentation:** 3
**Contribution:** 3
**Rating:** 6
**Confidence:** 4

**Summary:**

This paper aims to study the effect of data order in training pipeline that is later being unlearned could have significant impact on how unlearning might have effect. As such, they claim that ideal unlearning might be impossible as we don't have exact kwnoledge about the data order in training pipeline.

They further support this claim by an empirical setup where unlearning targets happen at different positions of training pipeline,  and observe that applying the same training procedure results in different outcomes, showcasing the senstitivy of unlearning algorithms on the place where unlearning targets are in training pipeline.

**Strengths:**

The idea of this paper is interesting. It analyzes how the exact position at which unlearning targets appear can affect unlearning performance, and thus why ideal unlearning may be impossible when the algorithm is not given this positional information.

The theoretical results look sound, and the empirical results support them.

Overall, the paper sheds light on a question that was not clearly studied before. Most prior work focused on cases where unlearning targets were introduced immediately before the unlearning step.

**Weaknesses:**

I think the main weakness is that the empirical results are somewhat limited. I would encourage the authors to run experiments on additional benchmarks such as RESTOR [1] and MUSE [2] to better assess the sensitivity of unlearning algorithms to how recently unlearning targets were introduced across different setups.

Also, I wouldn’t frame this as an impossibility of unlearning; rather, it is an impossibility of defining the ideal model when, in practice, we may not have access to it.

-------------


[1] Rezaei, Keivan, et al. "RESTOR: Knowledge Recovery in Machine Unlearning." arXiv preprint arXiv:2411.00204 (2024).

[2] Shi, Weijia, et al. "Muse: Machine unlearning six-way evaluation for language models." arXiv preprint arXiv:2407.06460 (2024).

**Questions:**

Given the results in this paper, what do the authors propose for evaluating machine unlearning in practical scenarios? In existing work, it is often assumed that the model was trained on the unlearning targets immediately prior to applying the unlearning algorithm, so the ideal model is simply the checkpoint before the introduction of those targets.

Do the authors suggest excluding all metrics that compare to this ideal model? Or do they offer ideas for redefining or approximating the ideal model so that such comparisons remain meaningful?

---

> ### Author Response · Authors · 2025-12-03
>
> We thank the reviewer for the positive assessment on the novelty and contribution of our work. The reviewer acknowledged that, by investigating staged training pipeliens widely used in LLMs today, our results bridge a critical gap in the current unlearning research community which focuses on unlearning being introduced immediately before the unlearning step. This is exactly why our experiments are *not* organized around existing peer-reviewed benchmarks.
>
> When a piece of forget data is learned in different orders, our results show that unlearning this data leads to path-dependent divergence. But what if the forget data is learned repeatedly throughout multiple training stages in different formats (e.g. common knowledge, news, textbooks)? How does permuting the order of *multiple* training stages, all of which contain the forget data, change unlearning behavior? This is an important open question yet outside of the scope of our work. The benchmarks proposed by the reviewer test unlearning ability precisely on these repeatedly-learned forget data, which is why our empirical results can not build on them.
>
> **Context on prior benchmarks (RESTOR and MUSE)**
>
> - the RESTOR benchmark studies *knowledge recovery*: its training datasets contain false information on common knowledge (e.g. “Nelson Mandela was born in *United States.* He was an *actor*…). After training on these contaminated knowledge, RESTOR evaluates the ability of models to recover its knowledge stage prior to learning the contaminated knowledge (i.e. being able to predict Nelson Mandela was born in South Africa).
> - the MUSE benchmark contains two unlearning data splits: Books and News. The Books split consists of the Harry Potter series. The News split consists of BBC news articles. The MUSE benchmark evaluates a model’s ability to forget instances from the aforementioned dataset splits while maintaining other desiderata such as utility preservation and privacy protection (under *membership inference attack*).
>
>
>
> Both benchmarks contain forget data that could have been repeatedly learned during the pre-training phase of base models subject to experiments.
>
> **(2) Framing: "Impossibility of Unlearning" vs. "Impossibility of Defining Ideal Model".**
>
> We strongly agree with your interpretation. This aligns perfectly with the core message of our paper.
>
> - We do not claim that *removing information* is impossible.
> - Rather, we claim that the **standard of Retrain Equivalence (RE)**—which assumes a single, unique "ideal model" exists regardless of history—is ill-posed.
> - As you noted, because we cannot know the training path in practice, we cannot define *which* retrained model is the correct target.

---

### Official Review · Reviewer_N2Pv · 2025-10-30

**Soundness:** 3
**Presentation:** 3
**Contribution:** 3
**Rating:** 6
**Confidence:** 3

**Summary:**

This paper investigates how the effectiveness of machine unlearning aimed at achieving Retain Equivalence varies depending on the order in which the model learns different datasets. The study focuses on local unlearning methods that rely solely on gradients computed from the forget set. The authors theoretically prove, under a linear model assumption, that the outcome of unlearning is path-dependent, meaning it can differ significantly depending on the sequence of training data. Empirically, the authors simulate a sequential post-training process using LLaMA and Qwen models and similarly observe that unlearning behavior is strongly path-dependent.

**Strengths:**

* Unlike prior unlearning studies that mainly focus on designing algorithms to maximize benchmark performance, this paper highlights a more fundamental issue: the effectiveness of unlearning critically depends not only on the algorithm itself but also on how the forget data was originally learned by the model.
* The argument is further strengthened by a theoretical result showing that, under a linear model assumption, performing local unlearning on two models trained with different data orders cannot simultaneously satisfy the defined notion of Retain Equivalence.
* Extensive experiments with LLMs empirically confirm the path-dependent nature of unlearning and show interesting phenomena such as the recency effect, thereby connecting the theoretical findings to observable behaviors in realistic post-training pipelines.

**Weaknesses:**

* Definition 2.1 formalizes Retain Equivalence (RE) as the pointwise closeness of model predictions on a test set (any generic test set $X_{test}$). While this assumption enables a theoretical analysis, it arguably over-constrains the notion of equivalence. In practice, two models trained on the same data but with different random seeds can yield noticeably different predictions, especially on out-of-distribution inputs, yet should still be regarded as equivalent from a statistical or functional standpoint. Therefore, RE as defined in this paper captures only pointwise similarity rather than distributional or behavioral equivalence. Extending RE to a distribution-level notion, such as expected loss or output distribution similarity under the retain data distribution, would make the theoretical conclusions more general and more aligned with standard unlearning objectives.
* Section 4.3 appears to overstate the evidence for “path-dependent superficial forgetting”. The claim of a newly identified phenomenon is supported by only 40 prompts for each dataset (C, U, and R), i.e., 120 samples total, in a largely qualitative setup, which is insufficient to draw reliable or generalizable conclusions about superficial vs. deep forgetting. While the observation is interesting as an illustrative example, it does not yet constitute empirical evidence of a new phenomenon.
* The paper defines local unlearning as using only the gradient of the forget set, thereby excluding many practical algorithms that employ lightweight retain-based regularization or contrastive objectives (e.g., Gradient Difference, DPO). This narrow definition simplifies the theoretical treatment but makes the conclusions less representative of real-world unlearning methods. As a result, the claimed impossibility result may not fully capture the feasibility of semi-local or weakly regularized unlearning approaches commonly adopted in practice.

**Questions:**

* Is there a specific reason for using different learning rates for both LLaMA2-13B and Qwen2.5-14B?
* Why did the authors choose to use LoRA? If the goal is to simulate the post-training process of LLMs, full fine-tuning might be a more appropriate setup. Also, during unlearning, are only the LoRA weights updated?
* In Figure 4, the results clearly show that unlearning is path-dependent, but how should we interpret this finding? In the setting where “deep forgetting” occurs, U is trained first. Does this imply that the effect arises because the recency effect is weaker in this configuration?

---

> ### Author Response · Authors · 2025-12-03
>
> We thank the reviewer for the positive assessment and for highlighting that our work addresses a fundamental issue in machine unlearning beyond simple benchmark performance. We fully address the reviewer’s identified weakness of our work in details below.
>
> **(1) Definition of Retrain Equivalence (RE) - Pointwise vs. Distributional.**
>
> We agree with the reviewer that strict pointwise equivalence is a high bar, given that even random seeds can cause notable unlearning outcome differences. However, since our results hold for *any* two models with different paths, our conclusions imply distributional divergence, assuming that training stages are randomly ordered. Please see our response to Reviewer q9sz for more details.
>
> **(2) Sample size in Section 4.3 (Superficial Forgetting).**
>
> We acknowledge that the initial sample size (120 samples) was limited due to the manual curation required to strictly adhere to safety and ethical guidelines regarding harmful content.
>
> We have since scaled up this experiment. By carefully leveraging base LLMs within a sandboxed environment (to prevent data leakage), we generated a larger dataset of 500 safety-sensitive prompts.
>
> The expanded experiments confirm our initial findings: the distinction between superficial and deep forgetting remains strongly path-dependent. We have updated Figure 4 and the corresponding section to reflect these robust results.
>
> (We have updated the Ethics Statement to reflect the use of base LLMs in generating harmful prompts. )
>
> **(3) Definition of Local Unlearning & Exclusion of Regularization.**
>
> We thank the reviewer for raising this important question. We were able to **expand our impossibility results to weakly local algorithms** both theoretically and empirically.
>
> **Summary of Extended Results**: We define weakly local algorithms to be those that contain retained-based regularization terms.
>
> - Theoretically, we proved that linear models trained with GA+Retained (a.k.a. Gradient Difference) also exhibit path-dependent divergence under mild assumptions.
> - Empirically, we ran the post-train → unlearn pipeline in Seciton 4 using two weakly local objectives, both of which are adopted in prior work (such as TOFU). Path-dependent divergence persists even with the precence of retained-based regularizers, which supports our theoretical results.
>
> The reviewers can see a more detailed discussion on weakly local unlearning in Appendix E and our response to reviewer q9sz.
>
> **(4) Questions.**
>
> - **Q1 (Learning Rates):** We used lower learning rates for larger models (Qwen2.5-14B) following standard scaling laws and empirical stability practices. Larger models typically require lower learning rate to maintain stable trainin. Since path-dependent divergence occurs in smaller learning rates of 5e-6, the divergence will only be more prominent with larger learning rates.
> - **Q2 (LoRA vs. Full FT):**
>
>     We utilized LoRA primarily for computational efficiency, as simulating multiple training permutations for entire post-training + unlearning pipeline is resource-intensive. Our implicit assumption is that LoRA should be a lower bound on the extend of path-dependent divergence, since models subject to full-parameter tuning has larger capacity to store path-dependent information.
>
>
> - **Q3 (Interpretation of Recency vs. Deep Forgetting):** This is a fascinating observation. In the "deep forgetting" case (where U is trained first), the model has seen many other tokens since learning U, potentially making the weights associated with U less "sharp" or "rigid" compared to recently learned data. This might allow the unlearning algorithm to erase the concept more broadly (deep forgetting) rather than just suppressing the specific logits (superficial). However, disentangling the specific causal mechanism between recency and forget-depth is complex, and we view this as a promising direction for future work.

---

### Official Review · Reviewer_q9sZ · 2025-11-05

**Soundness:** 3
**Presentation:** 3
**Contribution:** 2
**Rating:** 4
**Confidence:** 3

**Summary:**

This work considers unlearning within a multi-stage training framework. Specifically, this work propose a theoretical understanding that model's behavior during unlearning is influenced by the order of its training stages during learning. Therefore, it is impossible to universally achieve a retrained equivalent model with a path agnostic algorithm. Experimental results on different LLMs and unlearning heuristics verify the theory.

**Strengths:**

- The paper look at unlearning at a novel perspective: under multi-stage training.
- The paper propose a novel theoretical analysis for the impossibility result of universal relearning equivalence, which is important for understanding unlearning.

**Weaknesses:**

While I appreciate the theoretical insights provided by this work, I have a few concerns.
- How large is $t_0$? Looks like corollary 1 does not hold when $t* < t_0$. Then does that mean when using less amount of unlearning iterations, it is more likely that the model will achieve $\varepsilon$-RE?
- The theoretical results is based on 1. linear models, 2. gradient ascent. It's very different from the experiments which focuses on LLM, with different stages, and each stage will use some different method for learning. Grad ascent is also just a most naive unlearning heuristics for LLM.
- Given the theoretical analysis, I think it's more valuable to provide more simplified experiments to verify the theoretical understanding (e.g. classification tasks or even simpler linear models)
- I'm not convinced by the significance of the universality of retrain equivalence. Why do we need to find a universal $t$, such that under any path, the model unlearned at step $t$ has to achieve the same $\varepsilon$-RE? What's the issue with having different $t$ for different paths?

**Questions:**

- What is the motivation of using an approximate unlearning definition that measures similarity in output space, rather than the classic approximate unlearning definition which measures similarity in the model parameter distribution space?

---

> ### Author Response · Authors · 2025-12-03
>
> We thank the reviewer for the thoughtful feedback and for recognizing the novelty of our perspective on multi-stage training and our theoretical analysis. We fully address the reviewer’s raised concerns with our work in the following:
>
> **(1) How large is $t_0$? Does $t^\* < t_0$ imply easier Retrain Equivalence (RE)?**
>
> This is an insightful question. In our theoretical analysis, $t_0$ represents the threshold after which the exponential divergence dominates the constant term $C_0$ (See Appendix C).
>
> To empirically verify the magnitude of $t_0$, **we followed your suggestion to conduct simplified experiments** (see Point 2 below) using real-world datasets from OpenML. We observed that $t_0$ is often negligible (negative) in practice, meaning path-dependent divergence begins almost immediately. We have added the results and analysis in Appendix D.
>
> **(2) Request for simplified experiments to verify theory.**
>
> We appreciate this suggestion. We have added a **new section in the Appendix D** detailing experiments on linear regression tasks.
>
> **Setup:** We trained linear models on popular datasets from OpenML. For each task, we split the dataset into two partitions using k-means to ensure sufficiently distinct training stages. We fixed an error tolerance $\\epsilon = 0.1$ and observed the following results:
>
> | Dataset | $t^{\star}$ | $t_0$ | $\rho_{\star}$ | $\sigma_U$ | $\|P_U\Delta\theta_0\|_2$ | $\|(I-P_U)\Delta\theta_0\|_2$ |
> |--------------------|--------------|-------|------------------|-------------|--------------------------------|-----------------------------------|
> | diabetes | 19 | -299 | 0.107 | 0.747 | 0.276 | $2.577\times 10^{-16}$ |
> | california_housing| 11 | -78 | 0.501 | 9.499 | 0.096 | $6.477\times 10^{-17}$ |
> | linnerud_weight | 0 | -964 | 0.036 | 1.074 | 0.843 | $1.522\times 10^{-16}$ |
> | bike_sharing | 184 | -124 | 0.021 | 7.733 | 0.176 | $5.238\times 10^{-5}$ |
> | energy_y1 | 52 | -401 | 0.063 | 0.980 | 0.099 | $3.477\times 10^{-14}$ |
> | energy_y2 | 52 | -401 | 0.063 | 0.980 | 0.099 | $3.477\times 10^{-14}$ |
> | sgemm_gpu | 48 | -201 | 0.117 | 186.2 | 0.020 | $9.079\times 10^{-13}$ |
>
> **Analysis**
> - **Negative $t_0$ Indicates Immediate Divergence**: In all tested datasets, the calculated $t_0$ is negative. Recall that Theorem 3.1 predicts exponential divergence for all $t \ge \max(t_0, 0)$. A negative $t_0$ implies that the exponential mode dominates the constant error term from the very first step of unlearning. Thus, there is no “safe” period of unlearning where the path dependence is hidden; the divergence begins immediately.
> - **Nonzero Projection on Forget Span.** The term $\|P_U \Delta \theta_0\|_2$ is non-zero across all datasets. This empirically validates Lemma C.2 (non-orthogonality to the forget span), confirming that the initial weight difference due to training order is never perfectly orthogonal to the forget set, which underpins the exponential growth.
>
> **(3) Gap between Theory (Linear + GA) and Experiments (LLM + Complex Stages).**
>
> **Added Theoretical Results:** In light of the reviewer’s comment that gradient ascent is the most naive baseline for unlearning, we **expanded our theoretical results** as follows:
>
> - We proved that, for overparametrized linear models, weakly-local unlearning objectives of the form $\mathcal{L}(\theta) = \ell(\theta, X_{F}) - \ell(\theta, X_R)$ (i.e. adding a regularization erm on the retained set) also diverge exponentially.
> - Empirically, we validated that such non-local objectives also exhibit path-dependent divergence on Llama13B and Qwen14B. The results are as follows:
> | Path $p$ | Llama13B GA+Retain | Llama13B GA+KL | Qwen14B GA+Retain | Qwen14B GA+KL |
> |----------|---------------------|----------------|-------------------|----------------|
> | **Forget Score $\\uparrow$** |               |                    |                |                |
> | 1        | 0.816               | 0.447          | 0.562             | 0.065          |
> | 2        | 0                   | 0.073          | **1**                 | **1**              |
> | 3        | **1**                   | **1**              | 0.709             | 0.108          |
> | 4        | 0.121               | 0              | 0                 | 0              |
> | **Utility Score $S_{\\text{TOFU}}$ $\\uparrow$** | |                |                |                |
> | 1        | 1                   | 1              | 1                 | 0.798          |
> | 2        | **0.854**               | **0.904**          | **0.853**             | **1**              |
> | 3        | 0.528               | 0              | 0.562             | 0.551          |
> | 4        | 0                   | 0.393          | 0                 | 0              |
>
> For more details, the reviewer can visit Appendix E of our updated paper. We thank the reviewer for their feedback, which helped us strenghten our results.

---

> > ### Author Response · Authors · 2025-12-03
> >
> > **(4) Significance of "Universality": Why do we need a universal $t$?**
> >
> > We clarify here that the "universality" requirement stems from the definition of a **path-agnostic unlearning algorithm**.
> >
> > In realistic unlearning scenarios (e.g., removing toxic data from a downloaded model), the unlearning algorithm (the user) does not have access to the model's training history (the path). The algorithm only sees the current weights $\theta$ and the forget set $D_f$.
> >
> > - If we allow different $t$ for different paths, the algorithm would need an "oracle" that reveals the training history to select the correct $t$.
> > - Since local algorithms do not have this information, they must apply a **universal** rule (e.g., "run GA for 100 steps").
> > - Our theorem proves that *any* such universal rule will fail for at least some training paths. Thus, without knowledge of the path, reliable unlearning is impossible.
> >
> > **(5) Motivation for Output Space vs. Parameter Space definition.**
> >
> > Our choice of defining RE over model output space is motivated by the following:
> >
> > 1. **A Stronger Impossibility Result:** Convergence in parameter space implies convergence in output space. By contrapositive, divergence in output space implies divergence in parameter space. Since we prove that models diverge in output space, they necessarily fail to achieve RE in parameter space as well. Therefore, our impossibility result holds for the stricter parameter-space definition too.
> > 2. **Relevance to LLMs:** In deep learning, parameter space distance is often ill-defined due to parameter space symmetries (two models can have different weights but similar functions, see [1]). Output space similarity is the standard for assessing utility and safety in LLMs, making it the more practical metric for our primary experimental domain.
> >
> > In the following section, we develop how our main theoretical results can extend to the widely used $(\\epsilon,\\delta)$-certified unlearning over distributions in parameter space.
> >
> > **Overview**: $(\\epsilon,\\delta)$ unlearning also assumes that data points are trained in *i.i.d* batches. Their canonical definition does *not* work with staged training. In light of the reviewer’s suggestions, we modify $(\\epsilon,\\delta)$-unlearning to represent randomness in stage order and show that RE is still impossible in this definition, because
> >
> > **Setup**: Assume stage ordering is randomized. There are $N$ training stages, one of them to be unlearned. Training uses a random ordering $\\sigma$ over $N$ stages, retraining uses a random stage ordering $\\tau$ over $N-1$ stages.  Let $\\Theta$ be the parameter space; and let $f: \\Theta \\rightarrow \\mathbb{R}^{m \\times n}$  map parameters to output space.
> >
> > **Premise**: Two models trained with different stage ordering diverge from each other after $t$ steps of identical unlearning operations.
> >
> > **Distributional RE def.** Let $\\mathcal{W}$ be the parameter space shared by unlearned model $\\theta_u$ and retrained model $\\theta_r$. An $(\\epsilon,\\delta)$-unlearning holds when the following holds for any $W \\in \\mathcal{W}$.
> >
> > $$
> > \\begin{aligned}
> > & \\operatorname{Pr}\\big( \\theta_u \\in W ) \\leq e^\\epsilon \\, \\operatorname{Pr}\\big( \\theta_r \\in W ) + \\delta, \\quad \\text{and}\\\\
> > & \\operatorname{Pr}\\big( \\theta_r \\in W ) \\leq e^\\epsilon \\, \\operatorname{Pr}\\big( \\theta_u \\in W ) + \\delta
> > \\end{aligned}
> > $$
> >
> > **Proof sketch.** It suffices to find a counter-example of $W$. Let $d(\\cdot)$ be the output distance defined in paper. Define
> >
> > $$
> > W_r := \\biggr\\{ \\theta \\in \\Theta: \\min_{\\tau} d\\Big(f(\\theta), f(\\theta_r^{\\tau})\\Big) > r  \\biggr\\}
> > $$
> >
> > By construction, no retrain outcome lies in $W_r$.  Let
> >
> > $$
> > \\text{Good} := \\biggr\\{\\sigma: \\exists \\tau \\text{ s.t. }d\\big( f(\\theta_u^\\sigma), f(\\theta_r^\\tau ) \\big) \\leq r \\biggr\\}
> > $$
> >
> > Fix any retrain outcome $\\theta_r^{\\tau}$, our impossibility result implies that, for large enough $t$, *at most one* training order can be close to $\\theta_r^\\tau$. So there are at most $(N-1)!$ number of training histories that lie outside of $W_r$. That is,
> >
> > $$
> > \\text{Pr}(\\sigma \\in \\text{Good})\\leq \\frac{1}{N}
> > $$
> >
> > We can invoke the definition now. Note that $\\text{Pr}(\\theta_r \\in W_r) = 0$ by construction, leaving $\\delta \\geq 1 - \\frac{1}{N}$.
> >
> > [1] Lim, Derek, et al. "The empirical impact of neural parameter symmetries, or lack thereof." Advances in Neural Information Processing Systems 37 (2024)

---

### Author Response · Authors · 2025-12-03

We sincerely thank all reviewers for their constructive feedback and suggestions. We are delighted to see that the reviewers have a high appreciation of the importance of our worked setting of multi-staged training followed by unlearning, and the novelty and fundamental nature of our results on path-dependent divergence in LLM unlearning.

Two common concerns shared by reviewers are addressed in our revised draft:

1. **Expanded the Scope of Purely Local Algorithms**: Two reviewers identified the scope of purely local algorithms as the critical weakness of our work. Our revised draft expanded our theorem on  path-dependent divergence  also to **weakly local unlearning** algorithms, which contains regularization terms on retained data, since reviewers pointed out these are are widely adopted to stablize unlearning. Our results imply impossibility of Retrain Equivalence for this class of algorithms too.
2. **Corollary on the Distributional Definition of RE:** Two reviewers identified the point-wise definition of Retrain Equivalence as a weakness of our work. We address this by providing a walk-through how our point-wise divergence results imply violation of the probabilistic definition of RE.

For other concerns, we addressed them in our responses.

---

### Meta-Review · Area_Chair_oCLm · 2026-01-07

**Summary:**

The reviewers raised a consistent set of concerns around novelty, positioning, and conceptual clarity. While the paper offers a careful theoretical analysis of unlearning under staged training and provides useful empirical evidence of path dependence in LLM post-training, several reviewers questioned whether the central "impossibility" claim meaningfully extends beyond prior results on forgeability and ill-defined retrain equivalence. In particular, the lack of a clear, well-situated distinction from established unlearning theory weakens the contribution, even if the staged training perspective is interesting. Additional concerns include the reliance on linear-model theory to motivate conclusions for deep models, limited engagement with existing unlearning benchmarks, and framing choices that may overstate the scope of the result. Taken together, these issues suggest that the paper is not yet sufficiently well-positioned or differentiated to warrant acceptance in its current form.

**Reviewer Concerns:**

Concerns that were reasonably addressed: The rebuttal did a solid job responding to several technical questions. In particular, the authors clarified the role of staged training and why a universal, path-agnostic notion of retrain equivalence is ill-posed when training history is unknown. They also strengthened the paper by adding simplified linear-model experiments to validate the theory, expanding empirical results beyond the original setup, and extending the analysis from strictly local unlearning to weakly local objectives with retain-set regularization. Several reviewers' questions about the choice of output-space retrain equivalence, learning rates, LoRA usage, and the interpretation of recency effects were answered clearly and consistently with the paper's scope.

Concerns that remain outstanding: That said, a number of substantive issues were not fully resolved. Most importantly, the relationship to prior work on forgeability and auditable unlearning (e.g., Thudi et al., Kong et al.) remains insufficiently reconciled. While the rebuttal argues that the paper is "not about forgeability," the core impossibility result overlaps conceptually with earlier findings that retrain equivalence is ill-defined under limited information. The rebuttal does not fully convince that the novelty is clearly separated from this literature, nor does it sufficiently temper claims of conceptual originality. Additionally, some reviewers remain unconvinced by the framing of "impossibility of unlearning" rather than "impossibility of defining a unique ideal model," and by the strength of conclusions drawn from linear-model theory to complex LLM pipelines. Overall, while the rebuttal improves clarity and scope, it does not fully close the gap on novelty positioning and theoretical context.

**Reviewer Scores:**

Reviewer q9sZ: Likely be the same or a slight increase (4 or from 4 to 6). The added linear experiments, clarification of t_0, and explanation of universality directly address most of this reviewer's technical concerns, though some reservations about significance may remain.

Reviewer N2Pv: Likely remain the same (6). This reviewer was already broadly positive, and the expanded discussion on weakly local unlearning and larger empirical validation likely reinforced their confidence.

Reviewer QvRz: Likely remain the same (6). The rebuttal aligns with this reviewer's interpretation that the issue is about defining the ideal model rather than impossibility of unlearning, but does not fundamentally change their stance on empirical breadth.

Reviewer S5jh: Unlikely to increase meaningfully (likely remains at 2). Although the rebuttal acknowledges prior work and promises better citation and positioning, it does not fully accept or integrate the reviewer's core critique that much of the ill-posedness argument already exists in the forgeability literature. This reviewer explicitly framed acceptance as contingent on reframing, which remains incomplete.

---

### Decision · Program_Chairs · 2026-01-26

Reject